# Unique Phase Transition of Exogenous Fusion Elastin-like Polypeptides in the Solution Containing Polyethylene Glycol

**DOI:** 10.3390/ijms20143560

**Published:** 2019-07-20

**Authors:** Zhongqi Ge, Ziyang Xiong, Dandan Zhang, Xialan Li, Guangya Zhang

**Affiliations:** Department of Bioengineering and Biotechnology, Huaqiao University, Xiamen 361021, China

**Keywords:** SpyCatcher, PEG, phase transition temperature, elastin-like polypeptides

## Abstract

Elastin-Like polypeptides (ELPs), as well-known temperature-controlled bio-macromolecules, are widely used. However, little is known about the interactions between ELPs and macromolecules, which is an important yet neglected problem. Here, the phase transition characteristics of an ELPs-SpyCatcher fusion protein (E-C) in the presence of polyethylene glycol (PEG) in single salts (Na_2_CO_3_, Na_2_SO_4_, NaCl) solutions were investigated using a UV spectrophotometer, DLC, and fluorescence spectroscopy, and we got some interesting results. The phases transition of E-C occurred at a concentration lower than 0.5 mol/L Na_2_CO_3_/PEG2000, while in single Na_2_CO_3_ (<0.5 mol/L), the phase transition of E-C did not occur. In the Na_2_CO_3_/PEG solution, we observed a unique two-step phase transition of E-C when the Na_2_CO_3_ concentration was 0.5 mol/L and PEG2000 concentration was less than 0.15 g/mL, respectively. In the Na_2_CO_3_/PEG2000 solution, the phase-transition temperature of E-C decreased with the increase of PEG concentration, but increased in the Na_2_SO_4_/PEG2000 solution, while it remained unchanged in the NaCl/PEG2000 solution. However, the phase-transition temperature of the linear ELPs40 decreased under the same salts/PEG2000 solutions. We also addressed the possible molecular mechanism of the interesting results. In contrast to the current well-understood salts-ELPs interactions, this work provides some new insights into the interaction between the PEG-salts-ELPs in solution.

## 1. Introduction

Elastin-Like polypeptides (ELPs) are a class of stimuli-responsive biopolymers [1,2], which show an inverse phase transition temperature (Tt) [3,4,5]_._ They have been used for various applications including protein purification [6,7,8], tissue engineering [9], drug delivery [10], and so on. The Tt of ELPs relates its topological by incorporating foreign fusions [11], lengths and concentrations [3,12], as well as the external (such as types of salts and sugar molecules) [13,14,15].

Among them, salts are the most significant factors that effect ELPs Tt, which is consistent with the Hofmeister order, and the anions are stronger than the cations [16,17,18]. Cho and others [19] found the kosmotropic anions decreased the lower critical solution temperature (LCST) by polarizing interfacial water molecules while the chaotropic anions showed salting-in properties at low salt concentrations. As for incorporating foreign fusions, due to their hydrophilicity/hydrophobicity and aggregation characteristics, the ELPs incorporating foreign fusions have different phase transition characteristics. Christensen and others [20] fused 11 different foreign proteins with ELPs, and compared them with the linear ELPs, the Tt of the ELPs fusion protein increased or decreased in varying degrees. Recently, Howarth and coworkers [21,22] found that SpyTag and SpyCatcher could form a specific isopeptide bond between Asp-117 of Spy-Tag and Lys-31 of SpyCatcher spontaneously. It could combine some artificial ELPs with unconventional nonlinear topologies including circular, tadpole, star, and H-shaped, which enriched the structure of traditional linear ELPs [11]. Zhang and others [23] found a converse Hofmeister order of the SpyCathcher-ELPs_40_ (E-C), and showed that E-C had no phase transition in Na_2_CO_3_ (<0.5 mol/L), but it was obvious in the Na_2_SO_4_ solution with the same concentration. This might be related to the negative charge on the surface of SpyCatcher.

For ELPs, as a model of intrinsically disordered proteins [24], there are many reports about the influences of salts on the phase transition of them. However, studies on the interactions between ELPs and organic macromolecules have rarely been reported [15], as well as the mechanisms underlying the interactions between the aqueous solvent and ELPs. On the other hand, the interaction of polyethylene glycol (PEG) with biological macromolecules is an important study as many applications of them in various fields, such as biology, medicine, biomedical engineering and so on [25,26]. In addition, PEG as crowding agents, it could enrich the research of ELPs in a single salt solution.

In this paper, we systematically investigated the influences of PEG on the phase transition of E-C in the presence of different salts and obtained some interesting results. Firstly, the E-C underwent a phase transition at low Na_2_CO_3_ concentration only when PEG exited. Secondly, we discovered a unique two-step phase transition of E-C in the solution with 0.5 mol/L Na_2_CO_3_ and some specific concentrations of PEG2000 for the first time. Finally, the PEG had opposite effects on the Tt of E-C in the solution containing CO_3_^2−^ and SO_4_^2−^.

## 2. Results and Discussion

### 2.1. Influences of Na_2_CO_3_/PEG Concentration on the PhaseTransition of E-C

The influences of Na_2_CO_3_/PEG concentration on E-C phase transition are shown in Figure 1. As can be seen from it, when the Na_2_CO_3_ concentration was 0.3–0.5 mol/L, 25 μmol/L E-C underwent a phase transition in the presence of PEG2000 and PEG6000. In addition, under the same concentration of Na_2_CO_3_, the phase transition temperature of E-C (Tt_E-C_) gradually decreased with the increasing concentrations of PEG. However, under the same PEG concentration, Tt_E-C_ decreased with an increase of Na_2_CO_3_ concentration. In the experiment, we excluded the possibility of the phase transition of PEG itself, as shown in Appendix A.

In order to study the phase transition characteristics further, the particle sizes and distributions of E-C in 0.3 mol/L Na_2_CO_3_ solution with different concentrations of PEG2000 were measured. As can be seen from Figure 2, the average particle size ranged from 194.4 nm to 1393 nm after phase transformation of E-C in 0.3 mol/L Na_2_CO_3_ solution containing PEG2000. Therefore, it was confirmed that a phase transition occurred in E-C in this solution system. It has been reported that different types of ELPs can form uniform aggregates with varying diameters from 6 nm to 1602 nm [27]. When the salt solution or other agents were added, the particle sizes of the ELPs had different degrees of decline [28,29]. Our results are in accordance with them.

E-C underwent a phase transition when the concentrations of Na_2_CO_3_/PEG2000 solution were 0.3 mol/L and 0.4 mol/L, respectively. However, the phase transition did not occur with less than 0.5 mol/L Na_2_CO_3_ alone_._ Our explanations are listed below. The calculated pI of E-C is 4.69 (http://web.expasy.org/protparam). After adding Na_2_CO_3_/PEG2000 in PBS buffer, the pH shifted from 7.4 to about 11, E-C was occupied with a large number of negative charges, thus forming the spread of double-charged layer and the hydration layer around E-C. The thickness of the electric double layer and the hydration layer was positively correlated with the charged charge. The double-charged layer also protected the hydrogen bond formed between the water molecule and the E-C. When the CO_3_^2−^ concentration was less than 0.5 mol/L, the Na^+^ in Na_2_CO_3_ neutralized the negative charges, compressed the thickness of the electric double layer and weakened the mutual repulsive force between the E-C. At the same time, it was limited to CO_3_^2−^ polarized water molecules to weaken the hydrogen bond between E-C and water; therefore, E-C did not have phase transition phenomena with a low concentration of CO_3_^2−^ (≤0.5 mol/L) in the PBS buffer. On the other hand, with the addition of PEG2000 (as shown in Figure 1), due to the excluded volume effects of PEG2000 [30], the combined water on the E-C surface was robbed and the hydration layer around the E-C was destroyed by PEG2000. Simultaneously, there may be many more firm binding sites between E-C and PEG2000 [31]. It has been reported that the long-chain PEG interacted with lysozyme around six amino acid residues (Glu35, Arg61, Trp62, Arg73, Lys96, and Aspl0l) [32,33]. E-C also has the same amino acids and is rich in lysine (Lys). The common effects of the above two aspects lead to the phase transition of E-C.

In addition, with the increasing of Na_2_CO_3_ concentration, Na^+^ had neutralized the more negative charge of E-C, the electric double layer became thinner and the E-C repulsion became weaker. At the same time, the CO_3_^2−^ polarizing water molecules were enough to destroy the hydrogen bonds formed between water molecules and ELPs, and then E-C would have low transition temperature. With an increase of the PEG2000 concentration, the excluded volume effects were getting stronger. As for the particle size shown in Figure 2, E-C was easier to aggregate and the phase transition temperature decreased. However, we failed to observe the phase transition of E-C when only adding PEG into PBS (Appendix A). The possible reason is the large amount of Asp, Glu, Arg, Lys that exist at the surface of the SpyCather. More importantly, the numbers of Asp and Glu are more than Arg and Lys, thus, the surface of the SpyCatcher possess negative charges, which PEG cannot shield [23]. Besides, the excluded volume effects of PEG [30] were also not enough to trigger the E-C to aggregate. Therefore, the phase transition phenomena could not occur.

Comparing Figure 1 with our previous results about E-C in PBS buffer containing Na_2_CO_3_ [23], we observed the crowding reagent PEG2000 could significantly increase the Tt_E-C_. For example, in the PBS buffer containing Na_2_CO_3_, the Tt_E-C_ was less than 10 °C when Na_2_CO_3_ was 0.5 mol/L. In the 0.5 mol/L Na_2_CO_3_/PEG2000 system, when the concentration of PEG2000 was 0.02 g/mL, the Tt_E-C_ reached about 80. This was presumably due to the fact that PEG is a long-chain macro-molecules polymer, it not only has the effect of exclusion volume but also has a certain package protection effect on E-C, which resulted in the increase of Tt_E-C_ after addition of PEG into the solution.

Comparing Figure 1a,b, we could see PEG6000 had a greater impact than PEG2000 on the E-C phase transition temperature, although the transition temperature of E-C both correlated negatively with PEG6000/Na_2_CO_3_concentration. This was probably because of the strong exclusion effect of PEG6000 compare with that of PEG2000. Additionally, the binding stability of PEG6000 to E-C might be related with the fact that it had more binding sites with E-C [30], this agreed with the results of Wu on bovine serum albumin [32].

The increase of Na_2_CO_3_ concentrations from 0.3 to 0.5 mol/L, as suggested by the rise in fluorescence intensity of the solution (Figure 3a), and a clear blue shift occurred suggesting more hydrophobic regions of E-C exposure. The non-polarity increased with Na_2_CO_3_ concentration increasing to 0.5 mol/L, the hydrophobic interactions between E-C and Na_2_CO_3_ were plenty, and E-C aggregated and the phase transition occurred, which further confirmed the results of Zhang et al. [23]. This finding also confirms that the increase in salt concentration would cause ELPs to expose more hydrophobic regions and strengthen their hydrophobic interaction [20]. As shown in Figure 3b, the maximum fluorescence intensity of E-C changed from 264 to 450.8, while PEG2000 concentration increased from 0.20 g/mL to 0.30 g/mL in the Na_2_CO_3_/PEG2000 solution. However, the maximum fluorescence intensity of the system was still less than that of the 0.5 mol/L Na_2_CO_3_ system’s (Na_2_CO_3_ concentration when E-C can undergo phase transition).

Namely, when PEG2000 was added, the non-polarity of the hydrophobic region of E-C was less than the non-polar region of the hydrophobic region of E-C in the Na_2_CO_3_ system. When PEG2000 was added to the system, the hydrophobic interaction force of E-C and an interaction force of PEG2000 with E-C were weakened, resulting in the increase of phase transition temperature when PEG2000 was added. In addition, with the increasing of PEG2000 concentration, the fluorescence intensity enhanced, indicating that PEG concentration increases, resulting in the non-polar E-C hydrophobic region enhancement, and then the phase transition temperature of E-C was decreased.

### 2.2. A Unique Two-Step Phase Transition of E-C in Na_2_CO_3_/PEG Solution

In the Na_2_CO_3_/PEG2000 system, when the Na_2_CO_3_ concentration was 0.4 mol/L (Figure 4a), E-C had only one phase change. In addition, when Na_2_CO_3_ was increased to 0.5 mol/L (Figure 4b) and the PEG2000 concentration was ≥0.15 g/mL, only one phase transition occurred in E-C and its relative turbidity (*A_350_*) reached a maximum. However, when the PEG2000 concentration was less than 0.15 g/mL, E-C occurred concomitantly with the resolution of the phase transition into a low- and a high-temperature transition as illustrated by the two-step transition seen in 0.5 mol/L Na_2_CO_3._ The *A_350_* firstly sharply rose from 0.2 to 0.8 and was relatively stable over a certain temperature range. Then, the *A_350_* sharply increased again to a maximum of 1.4. Besides, the temperature range of the first step phase transition increased with the decrease of PEG2000 concentration. The phenomenon of such the two-step transition of ELPs has rarely been reported before. Although similar phenomena are occasionally reported in some other corresponding intelligent molecules like PNIPAM [34,35]. The possible reasons are listed below. When the PEG2000 concentration was less than 0.15 g/mL, Na^+^ in Na_2_CO_3_ neutralized the charge of E-C, when the CO_3_^2−^ concentration reached 0.5mol/L, compressing the thickness of its electric double layer, and the CO_3_^2−^ polarized water molecules, then the combined effects of the two were sufficient for the first phase transformation of E-C at low temperature. As the temperature increased, the interaction between E-C and PEG increased, prompting E-C to aggregate again, so the second phase transformation occurred.

### 2.3. PEG2000 had Opposite Effects on the E-C Phase Transition Temperature in the Presence of CO_3_^2−^ and SO_4_^2−^

We tested the Tt_E-C_ in Na_2_SO_4_/PEG2000 solutions and show the results in Figure 5. As can be seen, when the Na_2_SO_4_ concentration was 0.3 mol/L and 0.4 mol/L, the Tt_E-C_ was maintained at 26 °C and 19 °C with the increase of PEG2000 concentration. This indicates that the PEG2000 had almost no effect on the Tt_E-C_ in this condition. When the concentration of Na_2_SO_4_ increased to 0.45 mol/L and 0.5 mol/L, Tt_E-C_ increased from 17 °C to 27 °C (0.45 mol/L Na_2_SO_4_) with the increase of PEG2000 concentration, and from 10 °C to 31.2 °C (0.50 mol/L Na_2_SO_4_). This means the PEG2000 had great effects on the Tt of E-C when the concentration of Na_2_SO_4_ increased to 0.45 mol/L. As Na^+^ in Na_2_SO_4_ reduced the thickness of the electric double layer at low concentrations of Na_2_SO_4_ (0.3 mol/L and 0.4 mol/L), and its electric repulsion was weakened. However, because of PEG2000’s wrapping effect, when adding low Na_2_SO_4_ concentrations with PEG2000, Tt_E-C_ did not change much. At the higher Na_2_SO_4_ concentration (0.45 mol/L and 0.50 mol/L), when the PEG2000 concentration was increased, it was estimated that the encapsulation effects of E-C increased, resulting in an increased Tt_E-C_. The above analysis shows that the effect of PEG was different in various salt systems. In the Na_2_CO_3_/PEG2000 system, the PEG2000 exclusion effect dominated. In the Na_2_SO_4_/PEG2000 system, the PEG2000 encapsulation effect dominated.

To compare the effects of CO_3_^2−^ and SO_4_^2−^ on E-C, PEG2000 concentration versus phase transition temperature was plotted for different NaCl, Na_2_SO_4_ and Na_2_CO_3_ concentrations, as shown in Figure 6a,b. The results show that under the same PEG2000 concentration, Tt_E-C_ in the Na_2_CO_3_/PEG2000 system was higher than that of NaCl/PEG2000 and Na_2_SO_4_/PEG2000 under the same conditions; the highest difference was about 70 °C. As the experimental pH was 7.4, it shifted to 11 after adding Na_2_CO_3_, while it shifted to about pH 8 after adding NaCl and Na_2_SO_4_. Thus, the negative charge of E-C in Na_2_CO_3_/PEG2000 was obviously higher than that in NaCl/PEG2000 or Na_2_SO_4_/PEG2000, which caused E-C to have more mutual repulsion, resulting in the increase of Tt_E-C_. In addition, compared with two neutral solution systems, we found that Tt_E-C_ in the Na_2_SO_4_/ PEG2000 system was about 18 °C lower than that of NaCl/PEG2000 in 0.4 mol/L (Figure 6a). In addition, Tt_E-C_ did not increase with a PEG2000 concentration in 0.5 mol/L NaCl/PEG2000 (Figure 6b), which may be due to the special effect of SO_4_^2−^ on E-C. However, when E-C was in the solution with a lower Na_2_CO_3_ concentration (≤0.2 mol/L), the phase transition temperature could not be determined, as shown in Appendix A, the values of relative turbidity were near 0, which proved no phase transition occurred. Only when the S-shaped curve formed, it indicated the phase transition occurred.

We show the phase transition temperature of ELPs (Tt_ELPs40_) in a salt/PEG2000 system in Figure 6c,d. Tt_ELPs40_ in NaCl/PEG2000 system was higher than those of the other two salts/PEG2000, and the difference reached about 18 °C. Moreover, Tt_ELPs40_ in the Na_2_SO_4_/PEG2000 system decreased by about 3 °C with the increase of PEG2000 concentration. In the NaCl/PEG2000 system, Tt_ELPs40_ reduced by about 3–5°C with the increase of PEG2000 concentration. While in the Na_2_CO_3_/PEG2000, Tt_ELPs40_ was below 10 °C, but in the buffer containing 0.20 g/mL PEG2000 and 0.4 mol/L Na_2_CO_3_, the phase transition of ELPs40 occurred when the temperature was near 0 °C, the same as in the 0.5 mol/L Na_2_CO_3_/PEG2000 (Appendix A), which was also difficult for us to determine the phase transition temperature. Importantly, the Tt_E-C_ trends in the salt/PEG2000 system were different from ELPs40. This could due to the fused SpyCatcher spreading a large number of positive and negative charges on the surface, and the negative charge was more than the positive charge [24]. This could be the main factor that led to the different phase transition behavior of E-C in the different salts/PEG2000 solutions.

## 3. Materials and Methods

### 3.1. Plasmid Construction

*E.coli* BL21 (DE3) strains and plasmids such as pUC-19-ELPs_40_, pET-22b(+), pET-ELPs40 were preserved by our laboratory. The SpyCatcher gene was synthesized by Sangon Biotech Co., Ltd. (Shanghai, China) and sequentially cloned into a pET-ELPs40 vector. To reduce the mutual interference, a twelve-residue linker GSSGSGSGSGSG [36] was fused between ELPs and the SpyCatcher. The plasmid pET-SpyCatcher-ELPs_40_ (Appendix A) was transformed into *E.coil* strain BL21 (DE3) for expression after being verified.

### 3.2. Protein Expression and Purification

Expressions were started by preparing a 10 mL overnight starter culture from frozen stock in LB medium supplemented with 100 g/mL ampicillin at 200 rpm and 37 °C. Then, according to the volume, 1:100 vaccinate to TB medium shaken at 200 rpm and 37 °C to an *A_350_* between 0.5 and 0.6, at which protein expression was induced by adding 0.1 mM IPTG (Isopropyl β-D-thiogalactoside) and changed the temperature to 20 °C. The cells were harvested after 16 h by centrifugation for 10 min at 4 °C and 9000 ×*g*. Then, the cells were suspended in PBS buffer pH 7.4 stored at 4 °C, washing and centrifuging it again at the same temperature and speed. We resuspended the cells and disrupted by ultrasonication on ice (ultrasonic 2 s, the interval of 4 s, 10 min). Then the solution was centrifuged at 4 °C for 10 min at 13,400 ×*g* to separate the soluble protein from insoluble cell disruption solution. The soluble supernatant, which contained ELPs, was then purified using the ITC (inverse transition cycling) method.

For purification of the ELPs, we added 2.5 mol/L NaCl to the solution and kept it in a hot bath for 15 min at 40 °C to trigger the ELPs phase transition. Then, we used warm centrifugation of the protein solution for 10 min at room temperature with 12,000 rpm to pellet out the aggregated ELPs. The isolated pellet was resuspended in PBS buffer (pH 7.4) and kept in an ice water bath for 20 min, then it was centrifuged at 4 °C for 10 min at 13,400 ×*g*. This process was repeated two times to purify the ELPs.

### 3.3. Protein Characterization

The purified ELPs and E-C were firstly characterized by using 12% gradient Tris-Glycerol SDS-PAGE (Lonza, Basel, Switzerland) to confirm the purity (Appendix A) and the accurate molecular weight were determined by MALDI/TOF according to Reference [23]. The protein concentration was quantified using UV absorption at 280 nm on a Biomate3 (Thermo Scientific, Waltham, MA, USA). The extinction coefficient used to convert absorbance to concentration was calculated based on the tyrosine and tryptophan content of the peptides [37].

To find out the effects of PEG on Tt _ELPs_, we used an UV-visible spectrophotometer (Analytik Jena AG, Jena, Germany) equipped with a temperature-controlled cell (Varian, PaloAlto, CA, USA) which was set up with an average temperature ramp of 1 °C/min to analyze the samples. The temperature corresponding to the midpoint of the baseline and maximum absorbance of the 350 nm absorbance curve is Tt [6]. All the experiments we carried out used the same solution without ELPs or E-C as the control.

The average particle size and distribution of E-C in different concentrations of Na_2_CO_3_ and Na_2_SO_4_ solutions were determined by a nanometer laser particle size analyzer (Malvern Instruments, Worcestershire, England) [38]. The running parameters were scattering angle 173°, HeNe laser (633 nm) with an output power of 10 mW. The concentration of E-C in each sample was 25 μmol/L, filtered with a 0.22 μm PVDF syringe filter before loading and then stabilized for 5 min at the phase transition temperature of the sample being measured.

We used 1-anilinonaphthalene-8-sulfonic acid (ANS) to measure the PEG binding to E-C. ANS is a fluorescent dye whose fluorescence is greatly enhanced once binding to the hydrophobic residues (such as tryptophan) of a protein. ANS displays a characteristic blue shift in its fluorescence maximum from 515 to 475 nm when the ANS binding regions of non-polarity increase. During the folding or unfolding process of proteins, exposure and change of hydrophobic patches in any folded/unfolded protein can be characterized [26,32]. The ANS binding to E-C was followed by measuring fluorescent intensity using a spectrofluorometer (F4600, Tokyo, Japan) with an excitation wavelength of 375 nm and emission was scanned between 400 and 600 nm. The molar ratio of protein (E-C) to ANS was 1:100. All experiments were performed at room temperature with an incubation time of 15 min.

## 4. Conclusions

In summary, we investigated the effects of phase transition in a single salt (Na_2_CO_3_, Na_2_SO_4_, NaCl) solution system of E-C after adding crowding reagents, which is important for a better understanding of phase transition characteristics of ELPs in vivo. Collective data indicated that when the concentration of Na_2_CO_3_ was lower than 0.5 mol/L, Tt_E-C_ decreased gradually with the increasing PEG concentrations. When the concentration of Na_2_CO_3_ was 0.4 mol/L, the phase transition of E-C was observed only after the concentration of PEG was above 0.06 g/mL. Interestingly, when the concentration of Na_2_CO_3_ was 0.5 mol/L, E-C underwent a unique two-step phase transition when the concentration of PEG2000 was less than 0.15 g/mL. To the best of our knowledge, this is the first reported phenomenon of such a two-step phase transition of ELPs. This work offers new insights into a better understanding of the phase transition characteristics of ELPs in vivo.

## Figures and Tables

**Figure 1 ijms-20-03560-f001:**
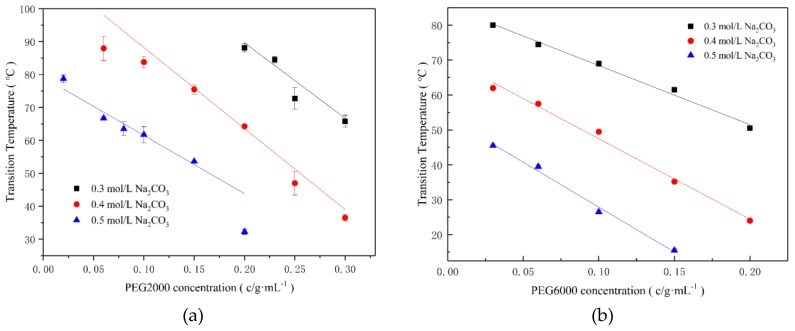
Effects of PEG/Na_2_CO_3_ concentration on E-C phase transitions temperature (Tt_E-C_). (**a**) PEG2000; (**b**) PEG6000.

**Figure 2 ijms-20-03560-f002:**
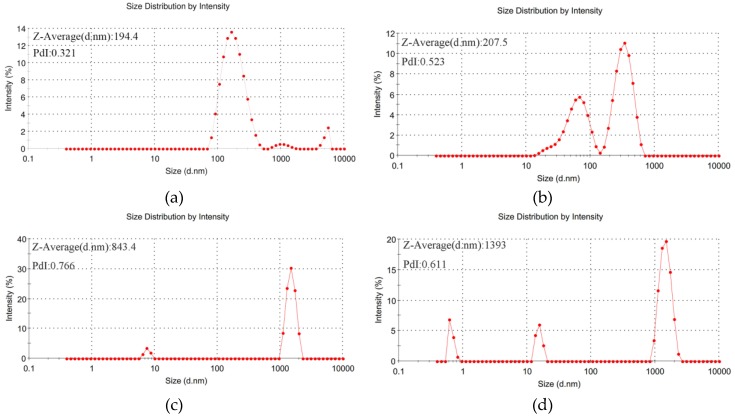
Particle size and distribution of 25 μmol/L E-C in 0.3 mol/L Na_2_CO_3_/PEG2000. (**a**) 0.20 g/mL PEG2000; (**b**) 0.23 g/mL PEG2000; (**c**) 0.25 g/mL PEG2000; (**d**) 0.30 g/mL PEG2000.

**Figure 3 ijms-20-03560-f003:**
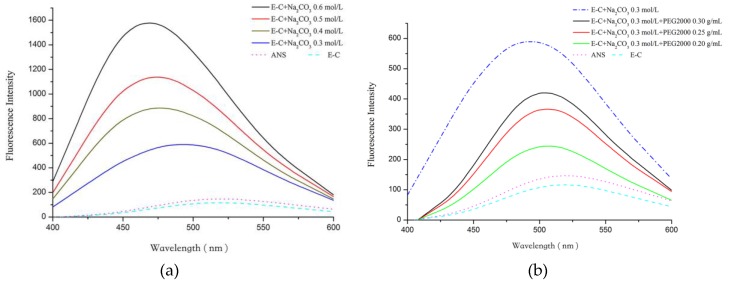
(**a**) ANS fluorescence characterization of Na_2_CO_3_–E-C interactions upon changes of Na_2_CO_3_ concentrations from 0.3 to 0.6 mol/L; (**b**) ANS fluorescence characterization of PEG2000/Na_2_CO_3_–E-C interactions upon.

**Figure 4 ijms-20-03560-f004:**
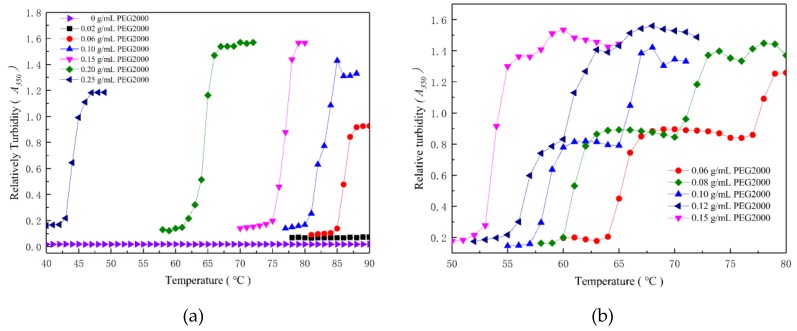
Effects of PEG2000/Na_2_CO_3_ concentration on 25 μmol/L E-C phase transitions temperature. (**a**) 0.4 mol/L Na_2_CO_3_; (**b**) 0.5 mol/L Na_2_CO_3._

**Figure 5 ijms-20-03560-f005:**
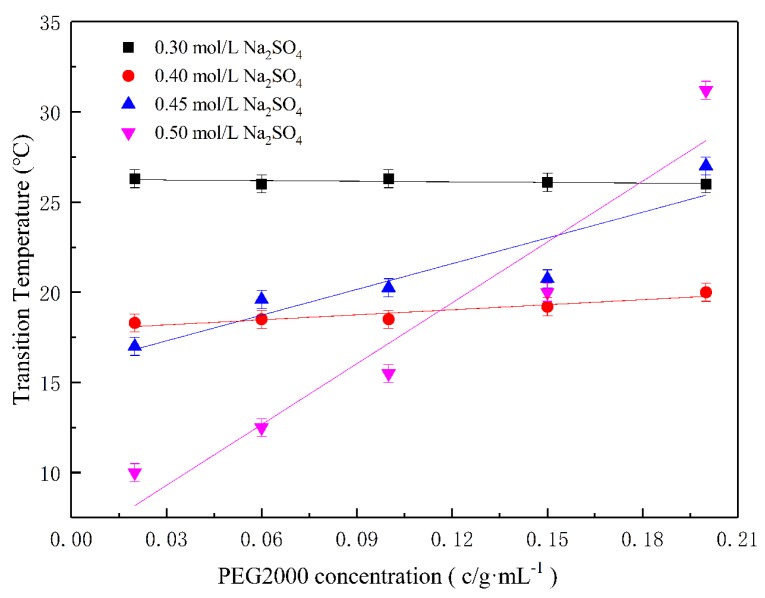
Effects of PEG2000/Na_2_SO_4_ concentration on the phase transitions temperature of E-C.

**Figure 6 ijms-20-03560-f006:**
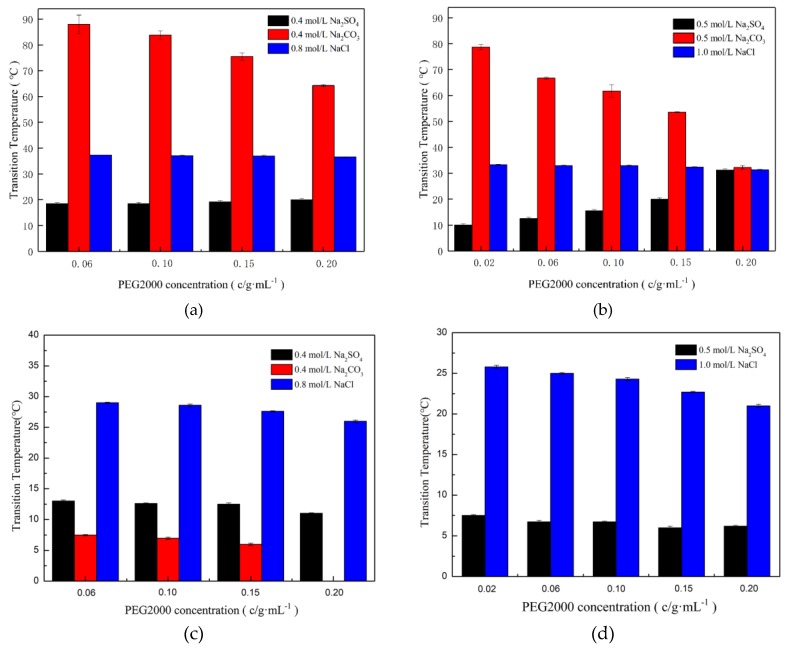
Effects of PEG2000 and salt concentrations on the phase transition temperature of E-C and ELPs_40_. (**a**) 0.4 mol/L salts + E-C; (**b**) 0.5 mol/L salts + E-C; (**c**) 0.4 mol/L salts + ELPs_40_; (**d**) 0.5 mol/L salts + ELPs_40_.

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
