# Peer review of "Unique Phase Transition of Exogenous Fusion Elastin-like Polypeptides in the Solution Containing Polyethylene Glycol"

_ijms, 2019, doi:10.3390/ijms20143560_

Round 1

Reviewer 1 Report

The manuscript is devoted to temperature dependent behavior of polypeptide/PEG/salt systems. I have serious doubts what is the origin of the detected phase transition. By my opinion the transition comes or from temperature sensitive PEG molecules either from combined transition of the both molecules. E.g., results showed in Fig. 1 are typical for PEG/salt solutions (see e.g. Boucher, E. A. and Hines, P. M. J. Polym. Sci., Part B: Polym. Phys. 1976, 14, 2241– 2251). I wonder that authors did not carry out experiments on PEG/salt solutions (without polypeptide) to distinguish thermal behavior of the polypeptide and PEG.

I could not support publishing of the manuscript in actual form. I recommend to perform additional experiments on PEG/salt solutions and on the basis of obtained results to revise results and their interpretation.

Author Response

After carefully read the paper the reviewer mentioned, we found the molecular weight of the polymer examined in the article of Boucher, E. A. and Hines, P. M. is 20000, while the molecular weight of the PEG we use was only 2000 and 6000. Possibly, due to the smaller molecular weight, we did not detect the phase transition of PEG in our experimental conditions and temperature range (Fig. S3) as no S-shaped curve observed. This indicated the phase transition of PEG had little effect on our experimental results. Besides, during the process of determination of ELPs phase transition, the samples
without ELPs were used as the control, and each point we have deducted the relative turbidity values of PEG (Fig. S3).

Reviewer 2 Report

The submitted manuscript “Unique Phase Transition of Exogenous Fusion Elastin-like Polypeptides in the solution containing Polyethylene Glycol” by Ge et al demonstrated a study of effects of phase transition in a single salt solution on Elastin-like Polypeptides (ELPs) or Spycathcher-ELPs40 (EA). Based on others’ methods, the authors expressed the ELP with spycathcher to form EA for this study. With the comparison of CO32-, SO42- and Cl-, they showed EA had phase transition at low CO32- concentration. Additionally, the transition temperature of EA decreased as the PEG concentration increasing, which showed the opposite effects on the transition temperature in the CO32- and SO42- solution. They also compared two kinds of PEG, including PEG2000 and PEG6000, which suggested both PEGs same effects on the EA transition temperature. These comparisons are interesting for further understanding of the phase transition characteristics of ELPs. However, their design of the experiments is the lack of comparisons. Therefore, I suggest it can be published on Int J Mol Sci after major revision.

The authors should address the comments below.

1. On page 2 line 71, they claimed “We failed to observe the phase transition of EA when only adding PEG into PBS.”, however, there was no supporting information to support this statement. Additionally, if it is possible, can they also explain this phenomenon?

2. On page 6 Figure 6, there were data missing in Figure 6 c and d. For Fig 6c, the data of Na2CO3 was missing at PEG2000 0.20 concentration. For Fig 6d, there was no data about Na2CO3 in this figure. They should add them in for a complete comparison.

3. For the supporting information, they should also provide the characterisation data of the expressed and purified proteins.

4. On page 2 line 69, the word “phrase” should be revised.

5. On page 3 line 98, they talked about the addition of PEG2000 into EA and salt solution. They should refer to Figure 1 in the main content here.

6. On page 3 line 107, they discussed the increasing of PEG2000 concentration for the particle size. So they should refer to Figure 2 in the content

7. On page 5 line 170, the word “indicate” should be revised.

8. On page 5 line 173, they discussed the effect of PEG2000 on the transition temperature of SO4-. Because they showed that effects of SO42- concentration 0.3/0.4 mol/l were not increasing. So, they should specify the SO42- concentration in this content.

9. On page 6 line 189, the number “8” should have “pH” before it.

10. On page 6 line 200, the word “Na2CO3” should be revised

Author Response

The submitted manuscript “Unique Phase Transition of Exogenous Fusion Elastin-like Polypeptides in the solution containing Polyethylene Glycol” by Ge et al demonstrated a study of effects of phase transition in a single salt solution on Elastin-like Polypeptides (ELPs) or Spycathcher-ELPs40 (E-C). Based on others’ methods, the authors expressed the ELP with spycathcher to form E-C for this study. With the comparison of CO32-, SO42- and Cl-, they showed E-C had phase transition at low CO32- concentration. Additionally, the transition temperature of E-C decreased as the PEG concentration increasing, which showed the opposite effects on the transition temperature in the CO32- and SO42- solution. They also compared two kinds of PEG, including PEG2000 and PEG6000, which suggested both PEGs same effects on the E-C transition temperature. These comparisons are interesting for further understanding of the phase transition characteristics of ELPs. However, their design of the experiments is the lack of comparisons. Therefore, I suggest it can be published on Int J Mol Sci after major revision.

-All the experiments we carried out used the same solution without ELPs or E-C as the control. Please see the details mentioned above. The authors should address the comments below.

1. On page 2 line 71, they claimed “We failed to observe the phase transition of E-C when only adding PEG into PBS.”, however, there was no supporting information to support this statement. Additionally, if it is possible, can they also explain this phenomenon?

-We have added the effect of adding only PEG to PBS on the phase transition of E-C (Fig. S4) and the explanation was also added. (Line 111 – 113)

2. On page 6 Figure 6, there were data missing in Figure 6c and d. For Fig 6c, the data of Na2CO3 was missing at PEG2000 0.20 concentration. For Fig 6d, there was no data about Na2CO3 in this figure. They should add them in for a complete comparison.

-The missing data is due to the fact that no phase transition occurs under this condition. As shown in Fig. S5, the values of relative turbidity were near 0, which proved no phase transition occurred. Only when the S-shaped curve formed, it indicated the phase transition occurred. We have already explained this in the supporting information (Fig. S5).

3. For the supporting information, they should also provide the characterisation data of the expressed and purified proteins.

-We added SDS-PAGE (Fig. S2) of the purified ELPs to the supporting information. The accurate molecular weight of the purified ELPs determined by the MALDI/TOF was shown above. As this figure has been published in the Chinese Journal of Chinese Science (reference 25), therefore, we did not add it in the support information. We have cited the reference instead. (Line 239-240)

4. On page 2 line 69, the word “phrase” should be revised.

-We have revised it.

5. On page 3 line 98, they talked about the addition of PEG2000 into E-C and salt solution. They should refer to Figure 1 in the main content here.

-We have revised it as the reviewer suggested. (Line 98)

6. On page 3 line 107, they discussed the increasing of PEG2000 concentration for the particle size. So they should refer to Figure 2 in the content.

-We have revised it as the reviewer suggested. (Line 109-110)

7. On page 5 line 170, the word “indicate” should be revised.

-We have revised it.

8. On page 5 line 173, they discussed the effect of PEG2000 on the transition temperature of SO4-. Because they showed that effects ofSO42-concentration 0.3/0.4 mol/l were not increasing. So, they should specify theSO42-concentration in this content.

-We have specified the concentration of SO42- in the manuscript. (Line 1754-1765)

9. On page 6 line 189, the number “8” should have “pH” before it.

-We have added pH before the number 8.

10. On page 6 line 200, the word “Na2CO3” should be revised.

-We have revised it.